# Transcriptome Analysis Revealed the Key Genes and Pathways Involved in Seed Germination of Maize Tolerant to Deep-Sowing

**DOI:** 10.3390/plants11030359

**Published:** 2022-01-28

**Authors:** Yang Wang, Jinna He, Haotian Ye, Mingquan Ding, Feiwang Xu, Rong Wu, Fucheng Zhao, Guangwu Zhao

**Affiliations:** 1The Key Laboratory of Quality Improvement of Agricultural Products of Zhejiang Province, College of Advanced Agricultural Science, Zhejiang Agriculture and Forestry University, Hangzhou 311300, China; wangyang@zafu.edu.cn (Y.W.); 2020601042009@stu.zafu.edu.cn (J.H.); 412591167@stu.zafu.edu.cn (H.Y.); 20110028@zafu.edu.cn (M.D.); 674456812@stu.zafu.edu.cn (F.X.); rongwu@stu.zafu.edu.cn (R.W.); 2Institute of Maize and Featured Upland Crops, Zhejiang Academy of Agricultural Sciences, Dongyang 322100, China

**Keywords:** deep sowing, maize seed germination, molecular mechanism, oxidative stress, plant hormone

## Abstract

To improve our understanding of the mechanism of maize seed germination under deep sowing, transcriptome sequencing and physiological metabolism analyses were performed using B73 embryos separated from ungerminated seeds (UG) or seeds germinated for 2 d at a depth of 2 cm (normal sowing, NS) or 20 cm (deep sowing, DS). Gene ontology (GO) analysis indicated that “response to oxidative stress” and “monolayer-surrounded lipid storage body” were the most significant GO terms in up- and down-regulated differentially expressed genes (DEGs) of DS. Kyoto Encyclopedia of Genes and Genomes (KEGG) analysis suggested that “phenylpropanoid biosynthesis” and “starch and sucrose metabolism” were critical processes in maize seed germination under deep-sowing conditions. Consistent with DEGs, the activities of superoxide dismutase, catalase, peroxidases and α-amylase, as well as the contents of gibberellin 4, indole acetic acid, zeatin and abscisic acid were significantly increased, while the jasmonic-acid level was dramatically reduced under deep-sowing stress. The expressions of six candidate genes were more significantly upregulated in B73 (deep-sowing-tolerant) than in Mo17 (deep-sowing-sensitive) at 20 cm sowing depth. These findings enrich our knowledge of the key biochemical pathways and genes regulating maize seed germination under deep-sowing conditions, which may help in the breeding of varieties tolerant to deep sowing.

## 1. Introduction

Maize (*Zea mays* L.) has the largest cultivation area and yield of all crops worldwide (http://faostat.fao.org/, 21 December 2021). In China, its planting area has exceeded 34 million hectares since 2012, representing the largest agricultural crop [1]. However, 2/3 of maize-cultivated land is located in arid and semi-arid regions, where drought is the main environmental stress factor affecting production [2,3]. Deep sowing is an effective strategy to overcome drought stress because seeds can absorb water from deeper soil layers to germinate normally [4]. Deep-sowing-tolerant varieties also reduce the sowing-failure risk caused by drought stress during seed germination in water-scarce environments. However, existing varieties show poor germination ability when seeds are sown into deeper soil layers [5]. Therefore, it is critical to develop deep-sowing-tolerant maize varieties specifically for arid and semi-arid regions. However, the genetic mechanisms and regulatory genes related to deep-sowing germination ability remain a subject of limited understanding and exploration.

In maize, Troyer [6] first demonstrated that deep-sowing tolerance of maize seeds was associated with mesocotyl elongation. Our previous study also drew a similar conclusion [7]. Several key genes or quantitative trait loci (QTL) related to deep-sowing tolerance were then identified and analyzed in our later research [1,3,8,9]. For instance, Affymetrix GeneChip analysis using a deep-sowing-tolerant maize inbred line, 3681-4, indicated that GA receptor GID1, transcriptional factor MYB, DELLA protein DWRF8 genes might play a critical role in GA3-induced mesocotyl elongation under deep-sowing conditions [4]. A *ZmMYB59* gene was then cloned, and its function was studied in detail by analysis of its expression at different sowing depths, germination stages, tissues and GA_3_ treatment. The results suggested that *ZmMYB59* played a negative regulatory role in seed germination under deep-sowing conditions [3]. Zhang et al. [8] identified 25 QTL related to deep-sowing tolerance across 3681-4 × X178 F_2:3_ families under both 10 and 20 cm depths and confirmed a major QTL for mesocotyl length. Liu et al. [1] also detected 8, 11, 13, 15 and 18 QTL for germination rate, seedling length, mesocotyl length, plumule length, and coleoptile length, respectively, under deep-sowing conditions from a high-resolution genetic map based on 280 lines of the intermated IBM Syn10 doubled-haploid population. Besides, six candidate genes related to deep-sowing germination ability were also identified. In another study, 243 IBM Syn4 recombinant inbred lines (RIL) constructed with B73 and Mo17 as parents were also used for QTL analysis of deep-sowing tolerance, and 7, 7, 5, 10 and 2 QTLs for emergence rate, seedling length, plumule length, mesocotyl length and coleoptile length, respectively, were detected. [9].

However, these studies have some limitations, such as low efficiency, lack of comprehensive understanding of the molecular basis of seeds germinated from deep soil layers, etc. To overcome these limitations, high-throughput sequencing technologies, such as RNA-Seq, provide an easy method of quick analysis of the transcriptomes of organisms with numerous genes [10], which makes it possible to better understand the molecular mechanism of seed germination in response to deep sowing and will aid in the breeding of deep-sowing-tolerant maize varieties. Zhao et al. [11] used RNA-sequencing to identify differentially expressed genes (DEGs) in both deep-seeding-tolerant W64A and intolerant K12 mesocotyls following culturing for 10 days after 2.0 mg∙L^−1^ 24-epibrassinolide (EBR) induced stress at the depths of 3 and 20 cm. The results showed that exogenous EBR, combined with deep-sowing treatment, triggered relevant signaling-transduction pathways, such as cell-wall-component and lignin biosynthesis, as well as the biosynthesis and signaling of phytohormones, then activated secondary and tertiary regulatory networks and finally altered the physiological characteristics of maize mesocotyls. Two-dimensional gel electrophoresis-based proteomic analysis was also used to reveal protein patterns during etiolated maize mesocotyl growth. The results indicated that a specific set of DAPs participate in various biological processes and underlie the cellular and physiological activities of the mesocotyl at different growth periods [12].

In our study, transcriptome analysis for deep-sowing germination ability of maize was performed utilizing the prevalent RNA-Seq technology. Seeds of maize inbred line B73 were used, soaked for 1 d, and embryos were separated from seeds germinated for 0 d (UG), NS (seeds germinated for 2 d at 2 cm sowing depth) and DS (seeds germinated for 2 d at 20 cm sowing depth). The results of RNA sequencing, real-time PCR validation of candidate genes and physiological metabolism analysis, including the determination of antioxidant enzyme activity, malondialdehyde and proline content, and endogenous phytohormone levels, provided novel insights into the molecular mechanisms of maize seed germination in response to deep sowing.

## 2. Results

### 2.1. Deep-Sowing-Induced Transcriptome Changes in Maize

The sequencing of maize embryo cDNA libraries generated, on average, 24,433,276 raw sequencing reads and 21,829,034 clean reads after removing post filtering of adapter sequences, as well as low-quality and short reads. Further, the clean reads were mapped to reference genome Zea_mays. AGP v3.22 using HISAT software. The average mapping ratio with the reference genome was 89.7%. The average proportion of uniquely and multiple-mapped reads was 79% and 10%, respectively.

DEG analysis showed that a total of 1349 DEGs were identified among maize embryos of UG, NS and DS, including 1029 upregulated DEGs and 320 downregulated DEGs (Figure 1). The cluster heat map of these DEGs is shown in Appendix A. Among the upregulated DEGs, 688 were identified between UG and DS, with 134 DEGs specifically detected in embryos of DS (Figure 1a). As for the downregulated DEGs, 93 were identified between UG and DS, with 47 DEGs specifically detected in embryos of DS (Figure 1b). The detailed information of all detected DEGs is listed in Appendix A.

### 2.2. Identification and GO Functional Enrichment Analysis of DEGs

DEGs between UG and DS and the specifically identified DEGs in embryos of DS were used to map the GO database to explore the significantly enriched terms compared with the genome background using AgriGO v2.0 toolkit, with false discovery rate (FDR values < 0.05) as the threshold. The results showed that a total of 28 significant GO terms were identified among three gene-ontology in the upregulated DEGs between UG and DS (Figure 2a). The main classes contributing to ‘biological process’ were “chromatin assembly”, “protein-complex assembly”, “response to cold”, “response to oxidative stress” and “movement of cell or subcellular component”. “Nucleosome”, “apoplast” and “microtubule-associated complex” were the enriched GO terms in the cellular-component category. In the molecular-function category, the primary classes were “electron-carrier activity”, “peroxidase activity”, “heme binding”, “copper-ion binding”, “hydrolase activity” and “ion transmembrane-transporter activity”. In the downregulated DEGs between UG and DS, only five significant GO terms were found, with “monolayer-surrounded lipid storage body” and “lipid particle” in the biological-process category; and “nucleic-acid binding transcription-factor activity”, “transcription-factor activity”, “sequence-specific DNA binding” and “signal-transducer activity” in the cellular-component category (Figure 2b).

In this study, we focused our attention mainly on DEGs detected only in embryos of DS; therefore, the significant GO terms of up- and downregulated DEGs of DS were also analyzed (Figure 2c,d). Among the upregulated DEGs, seven enriched GO terms were found; “response to oxidative stress” was the significant GO term in the biological-process category, whereas the remaining six GO terms all belonged to the molecular-function category, including “electron-carrier activity”, “heme binding”, “calcium-ion binding”, “peroxidase activity”, “monooxygenase activity”, “hydrolase activity, hydrolyzing O-glycosyl compounds” (Figure 2c). As for the downregulated DEGs, four significant GO terms were enriched, with “monolayer-surrounded lipid storage body” and “lipid particle” in the biological-process category and “nucleic-acid binding transcription-factor activity”, “transcription-factor activity” and “sequence-specific DNA binding” in the cellular-component category (Figure 2d).

### 2.3. KEGG Functional-Enrichment Analysis of DEGs

In order to elucidate the specific biochemical pathways during maize seed germination in response to deep sowing, DEGs between UG and DS and the specifically identified DEGs in embryos of DS were further searched against the KEGG pathway database. “Phenylpropanoid biosynthesis”, “biosynthesis of secondary metabolites”, “metabolic pathways”, “starch and sucrose metabolism”, “biotin metabolism”, “cysteine and methionine metabolism”, “benzoxazinoid biosynthesis”, “tyrosine metabolism”, “flavonoid biosynthesis” and “butanoate metabolism” constitute major pathways upregulated between UG and DS (Figure 3a). As for those genes only identified in embryos of DS, four pathways were significantly enriched, i.e., “phenylpropanoid biosynthesis”, “starch and sucrose metabolism”, “photosynthesis” and “ubiquinone and other terpenoid-quinone biosynthesis” (Figure 3b). No significantly enriched KEGG pathway was found in down-regulated DEGs between DS and UG or in those specifically identified in embryos of DS, according to the selection criteria *q* value < 0.05.

To propose a pathway involved in the process of maize seed germination under deep sowing, we first highlighted the DEGs in the phenylpropanoid biosynthesis pathway based on the KEGG pathway annotation from those upregulated DEGs specifically detected in embryos of DS (Figure 4). This identified some key genes of phenylpropanoid biosynthesis, including F5H (ferulate-5-hydroxylase, *AC210173.4_FG005*), 1.1.1.195 (cinnamyl-alcohol dehydrogenase, *GRMZM2G700188*), 3.2.1.21 (beta-glucosidase, *GRMZM5G810727*) and 1.11.1.7 (peroxidase, *GRMZM2G405459*). Their final products are coumarinate, p-hydroxy-phenyl lignin, guaiacyl lignin, 5-hydroxy-guaiacyl lignin and syringyl lignin. In addition, based on the KEGG pathway annotation, the expression profile of candidate genes in most represented pathways of “phenylpropanoid biosynthesis” and “starch and sucrose metabolism” are shown in Figure 5a,b, among samples of UG, DS and NS. Therefore, those candidate genes that specifically respond to deep-sowing conditions can be selected.

### 2.4. Experimental Validation of Differential Expressed Genes by qRT-PCR

qRT-PCR was performed on twelve genes in order to validate the differential gene expression obtained by RNA-Seq, including six upregulated DEGs (Figure 6a) and six downregulated DEGs specifically detected in embryos of DS (Figure 6b). These DEGs included candidate genes in the enriched KEGG pathways of “phenylpropanoid biosynthesis” and “starch and sucrose metabolism”, as well as enriched GO terms and genes related to phytohormones. The six upregulated DEGs were *GRMZM2G405459* (encoded a peroxidase protein), *GRMZM2G030790* (encoded a jasmonate-induced protein), *GRMZM5G809195* (encoded an IAA14 protein), *GRMZM2G078465* (encoded an indole-3-acetate beta-glucosyltransferase), *GRMZM2G103055* (encoded an alpha-amylase precursor) and *GRMZM2G093286* (tpa: cytochrome p450 superfamily protein). The six downregulated DEGs were *GRMZM2G043338* (encoded an auxin-repressed protein), *GRMZM2G067743* (encoded putative-growth-regulating factor 11), *GRMZM2G337229* (encoded a 16kda oleosin), *GRMZM2G096435* (encoded an oleosin bn-v), *GRMZM2G480954* (encoded a 18kda oleosin) and *GRMZM2G168474* (tpa: cis-zeatin o-glucosyltransferase partial). The results indicated that the expression pattern as obtained by qRT-PCR corroborated that obtained by RNA-Seq data for all twelve genes (Figure 6a,b). A Pearson correlation analysis between the gene-expression levels measured by qRT-PCR and RNA-Seq showed a significant correlation (correlation coefficient, R = 0.978) supporting the reliability of sequencing results (Figure 6c).

### 2.5. Deep-Sowing-Induced Changes in Phytohormone Levels in Maize

As phytohormones play important roles during seed germination and some DEGs (such as *GRMZM2G030790*, *GRMZM5G809195* and *GRMZM2G043338*) identified in embryos of DS were involved in phytohormone metabolism, we further compared the phytohormone contents in maize embryos among UG, NS and DS. After 2 d of germination, GA_1_ and GA_4_ content in maize embryos of both NS and DS increased significantly, as compared to that of UG (Figure 7a). GA_1_ content of NS and DS was 13.2 times and 12.1 times as much as that of the control, respectively. Compared with NS, GA_4_ content of DS increased by 27.7%, while there was no significant difference in GA_1_ content between NS and DS. However, GA_3_, which was considered to be the most important gibberellin during seed germination, was not detected in our study. IAA and zeatin contents were significantly higher in embryos of UG than those of NS and DS (Figure 7b). IAA content of UG was 7.37 times and 6.03 times as much as that of NS and DS, respectively. Compared with UG, zeatin content of NS and DS was reduced by 82.1% and 77.7%, respectively. Moreover, IAA and zeatin contents were more significantly accumulated in embryos of DS than those of NS, which consisted of the up-expression of *GRMZM5G809195* (encoded an IAA14 protein) and down-expression of *GRMZM2G043338* (encoded an auxin-repressed protein). JA and ABA were also determined in this research (Figure 7c,d). Our results showed that significantly more JA was accumulated in maize embryos of NS and DS, with contents 30.1 times and 12.4 times as much as those of UG, respectively. However, when compared with NS, JA content decreased significantly in embryos of DS (Figure 7c). There was no significant difference in ABA content between UG and NS. However, ABA content in embryos of DS was 5.29 times and 5.60 times higher than that of UG and NS, respectively (Figure 7d). These results indicated that endogenous phytohormone GA_4_, IAA, zeatin and ABA may play positive roles in maize seed germination in response to deep sowing, while JA negatively regulated seed germination under deep-sowing conditions.

### 2.6. Change in Enzyme Activity, Malondialdehyde and Proline Content of Maize in Response to Deep Sowing

In order to investigate the physiological changes in maize embryos in response to deep sowing, we measured the activities of several antioxidant enzymes, proline content, α-amylase activity and MDA content in embryos of DS, NS and UG. As expected, SOD, CAT and POD activities of DS were all significantly higher than those of NS, with activity was increased by 22.7%, 42.5% and 44.4%, respectively (Table 1). Compared with UG, the activities of these enzymes of NS were increased significantly, with SOD, CAT and POD values increased by 17.2%, 32.9% and 50.0%, respectively. Changes in MDA content showed the same pattern as those antioxidant enzymes among UG, NS and DS (Table 1). In addition, proline content of DS also increased significantly, with a value 2.31 times greater than that of NS. However, there was no significant difference in proline content between UG and NS. The activity of α-amylase in maize embryos of DS was significantly higher than that in NS or UG; besides, the activity of NS was also significantly higher than that of UG. In detail, the activity of NS and DS was 5.20 times and 9.84 times as much as that of UG, respectively. Moreover, we found that some physiological changes were consistent with the changes in gene expression, such as the up-expression of *GRMZM2G405459* (encoded a peroxidase protein), *GRMZM2G103055* (encoded an alpha-amylase precursor) and *GRMZM2G150893* (encoded a peroxidase precursor).

### 2.7. Expression Analysis of Candidate Genes of Different Deep-Sowing-Tolerant Maize Inbred Lines

According to the results of the above bioinformatics analysis, six candidate genes were selected to investigate the correlation between their expression and maize seed deep-sowing tolerance. The results showed that deep sowing (20 cm sowing depth) promoted upregulated expression of all six genes in both Mo17 (deep-sowing-sensitive) and B73 (deep-sowing-tolerant), except that *GRMZM2G043338* (encoded an auxin-repressed protein) and *GRMZM2G337229* (encoded a 16kda oleosin) were downregulated in Mo17 (Table 2). The relative gene-expression values of all six genes in B73 were significantly higher than those of Mo17, except for *GRMZM2G030790* (encoded a jasmonate-induced protein). Notably, deep sowing significantly upregulated the expression of *GRMZM5G809195* (encoded an IAA14 protein) and *GRMZM2G405459* (encoded a peroxidase protein) in B73, as compared with those of Mo17. Therefore, these four genes, i.e., *GRMZM5G809195*, *GRMZM2G405459*, *GRMZM2G043338* and *GRMZM2G337229* were worthy of further study.

## 3. Discussion

Deep sowing is a traditional and effective sowing method to ensure that seeds absorb water from the deep soil layer and germinate normally in water-scarce environments. Besides, deep sowing also prevents damage from the surface application of harmful chemicals [13]. Seedlings that emerge from deep soil should grow up faster than those emerging from shallow soil during dry spells, as the deeper seeds can still access stored soil water to support germination and emergence [14]. In addition, although deep sowing of peanuts resulted in poor root development and lower-leaf area index at the early growth stage, deep sowing not only increased root density, leaf-area index and total dry weight at 60 and 90 days after seeding but also increased seed yield and 100-seed weight [15]. Moreover, deep sowing also decreased the damage done by mice in southern Australia [16] and diminished the negative effects of parasitic weed [17]. Planting maize deep into stored soil moisture has been practiced in (semi-) arid regions to promote normal seed germination [1]. However, when sown deeply, the juvenile plant needs to elongate its organs to push the plumule to the soil surface [8]. Existing crop varieties, such as maize, showed poor emergence and weak seedling establishment when sowed deeply, which greatly restricts the potential application of deep-sowing technology.

To develop deep-sowing-tolerant maize varieties, especially for use in arid and semi-arid regions, it is important to understand the molecular basis of seed germination in response to deep sowing and explore genes related to deep-sowing tolerance. However, there are only a few studies in the literature that contribute to the understanding of the deep-sowing-tolerance mechanism in maize, which is basically explained by mesocotyl elongation and hormone response. To fill this knowledge gap, the present study sequenced and comparatively analyzed the transcriptome of maize embryos separated from seeds germinated for 0 d (UG), seeds germinated for 2 d at 2 cm sowing depth (NS) and seeds germinated for 2 d at 20 cm sowing depth (DS) with RNA-Seq technology. The number of DEGs detected in NS was over three times as much as that of DS, not including commonly identified DEGs (Figure 1). Therefore, it seems that physiological and biochemical processes may respond more slowly to deep-sowing conditions when compared with normal sowing. Another explanation is that when seeds were sowed in deep soil, the expression of genes related to stress resistance and energy metabolism were improved, whereas the expression of “unrelated redundant genes” were not induced. In fact, a series of subsequent GO, KEGG, physiological and biochemical analyses showed that the latter explanation may be more reasonable (Figure 2, Figure 3 and Figure 7; Table 1). The reliability of the RNA-Seq data was further verified by qRT-PCR analysis of the relative expression level of 12 selected DEGs (Figure 6).

In order to reveal the molecular mechanism of maize seed germination under deep-sowing conditions, we mainly focused on DEGs that were only detected under deep-sowing conditions, i.e., the 134 upregulated and 47 downregulated DEGs. GO enrichment analysis showed that these upregulated DEGs were mainly associated with defense response to oxidative stress, response to phytohormone stimulus and oxidoreductase activity, while the downregulated DEGs were associated with monolayer-surrounded lipid storage body and lipid particle (Figure 2c,d). These results indicated that deep-sowing had induced a series of environmental stress responses during maize seed germination, such as changes in phytohormone contents, antioxidant enzyme activities, etc. Subsequent experimental results supported this assumption (Table 1 and Figure 7). KEGG enrichment analysis indicated that “phenylpropanoid biosynthesis” and “starch and sucrose metabolism” were the significantly upregulated biochemical pathways during maize seed germinated under deep-sowing conditions (Figure 3b). Some key genes of phenylpropanoid biosynthesis were identified in our study (Figure 4); their final products were p-hydroxy-phenyl lignin, guaiacyl lignin, 5-hydroxy-guaiacyl lignin and syringyl lignin. These results are consistent with those of Zhao et al. [11], who also identified multiple DEG-encoding lignin biosynthesis pathways. As a major component of the secondary cell wall, lignin was found to be related to various stress-resistance responses of plants. In the mesocotyl of chilling-acclimated seedlings, lignin content was elevated to improve the mechanical strength of the mesocotyl [18]. Low lignin content was associated with low field emergence in unpigmented kabuli chickpeas [19]. Uniconazole significantly alleviated lodging stress by enhancing lignin biosynthesis and optimizing the culm morphological characteristics [20]. *ZmmiR528*- overexpressing transgenic maize plants had reduced lignin content and rind-penetrometer resistance and were prone to lodging under N-luxury conditions [21]. Besides, appropriate dosages of nanostructured lignin microparticles can be used to improve maize seed germination and radicle length [22]. In previous research, we also found several genes involved in cell-wall synthesis and cell elongation under deep sowing, which may also relate to the lignin metabolism [1,4]. In this study, the expression of one lignin biosynthesis gene, i.e., GRMZM2G405459, was increased more in embryos of a deep-sowing-tolerant maize inbred line (B73) than in those of a deep-sowing-sensitive line (Mo17) under deep-sowing conditions (Table 2). Therefore, our results suggest that the increase in lignin content plays an important role in maize seed germination under deep sowing conditions.

When seeds are deeply sown, more energy is needed to help the embryo break through the soil and germinate. Starch degradation induced by α-amylase is essential for the initial germination of cereal grains, including maize seeds [23]. The second significantly upregulated biochemical pathway found in maize embryo under deep sowing was “starch and sucrose metabolism” (Figure 3b). “Monolayer-surrounded lipid storage body” and “lipid particle” were the significantly downregulated enriched GO terms (Figure 2d). These results are consistent with previous studies on material decomposition and utilization during seed germination, such as seed-storage starch, protein, lipid, etc. [4,24]. Genes involved in “starch and sucrose metabolism” and “monolayer-surrounded lipid storage body” were selected and verified by RT-PCR, and the expression of *GRMZM2G103055* (an alpha-amylase precursor) was improved, whereas the expression of *GRMZM2G337229* (a 16kda oleosin), *GRMZM2G096435* (an oleosin bn-v) and *GRMZM2G480954* (a 18kda oleosin) was reduced in maize embryos under deep-sowing conditions (Figure 6). Besides, the activity of α-amylase was also found to increase significantly in maize embryos of DS (Table 1). Therefore our results confirm that more energy consumption is required when maize seeds are deeply sown.

Antioxidant enzymes, such as SOD, CAT and POD, are recognized as important scavengers of reactive oxygen species (ROS) in plants, and their activities could be increased when plants are subjected to stress conditions. In this study, the activities of SOD, CAT, POD and α-amylase increased significantly in maize embryos of DS, as did proline content, which is an important osmotic regulator in plants [25]. These results are consistent with earlier reports on lentil seed germination under osmotic stress [26], Brassica napus seed germination in response to salt stress [27] and wheat seed germination under low-temperature conditions [28]. Additionally, GO enrichment analysis also indicated that “response to oxidative stress” and “peroxidase activity” were the significant GO terms in DEGs detected only in embryos of DS, and expression of the selected gene-encoding peroxidase protein (*GRMZM2G405459*) was verified by RT-PCR and determined in both Mo17 and B73. Collectively, these results demonstrated that deep-sowing induced the stress-response mechanism of maize seeds.

Our previous studies showed that deep-sowing tolerance is associated with mesocotyl elongation [7], whereas phytohormones promotes germinability of maize in response to deep sowing [5,29,30]. Therefore, we measured several phytohormone contents in maize seed embryos (Figure 7). The results showed that GA_4_, IAA, zeatin and ABA contents in embryos of DS were higher than those of NS, while JA content was lower than that of NS. It was speculated that GA might have induced the synthesis of α-amylase, which promoted the degradation of starch and the production of energy. Changes in GA_4_ and IAA contents were consistent with previous studies [5,29]. Hu et al. [31] found that mesocotyl elongation of rice was associated with accumulated cytokinin content, and Feng et al. [32] suggested that light inhibition of mesocotyl elongation in rice could be caused by both lower functioning of growth-enhancing phytohormones (IAA, t zeatin, GA_3_) and higher levels of repressing phytohormone (JA). In addition, Xiong et al. [33] also indicated that the reduction in JA levels enhanced the growth of mesocotyl and coleoptiles of rice. The current research indicates that JA may play a negative regulatory role during seed germination. Moreover, promotion of ABA on growth of rice mesocotyls was previously demonstrated by Watanabe et al. [34]. Our results are in accordance with these previous reports. Moreover, DEGs related to phytohormone metabolism were detected in the present study, which is consistent with our earlier research [1,4]. The expression profile of these genes was also verified and investigated in deep-sown embryos of Mo17 and B73. The expression changes of these genes were in line with changes in the levels of phytohormone. For instance, *GRMZM5G809195* encoded an IAA14 protein, which is a member of the Aux/IAA protein family, involved in the regulation of lateral root development [35,36]. *GRMZM2G078465* encoded an indole-3-acetate beta-glucosyltransferase, which catalyzes the conjugation of IAA with glucose, contributing to the homeostasis of phytohormone in plant [37]. In this study, greater IAA content of DS may be related to the upregulation of these two genes, which may also suggest that deep-sowing-induced IAA accumulation is necessary for the rapid elongation of maize embryos. Therefore, our data again indicate that phytohormones regulated the elongation of mesocotyl or seed germination by acting antagonistically or complementarily in the hormonal interaction [31,33].

## 4. Materials and Methods

### 4.1. Preparation of Plant Material

Seeds of maize inbred line B73 were used in this study. Dry seeds were surface-sterilized with 0.5% sodium hypochlorite solution for 5 min, followed by thorough washing with water. Then, seeds were soaked in tap water for 1 d at 25 °C. Next, seeds were sowed in moist sand with a depth of 2 or 20 cm. Each treatment contained three repetitions, with 100 seeds per repeat. All seeds were germinated in growth chambers with a photosynthetic photon flux density (PPFD) of 250 μmol m^−2^s^−1^ and a 12-h photoperiod at 25 °C. Embryos were rapidly separated from seeds germinated for 0 d (UG), germinated for 2 d at a 2 cm sowing depth (NS) and germinated for 2 d at a 20 cm sowing depth (DS), respectively. Then, embryos were used for metabolite analysis or frozen in liquid nitrogen and stored at −80 °C for RNA extraction.

### 4.2. Measurements of Enzyme Activity, Malondialdehyde and Proline Content

For antioxidant enzyme activities and determination of malondialdehyde (MDA) content, about 0.15 g (fresh weight, FW) of embryos was ground in 4 mL of 0.05 mol/L sodium-phosphate buffer (pH7.8) and centrifugated at 10,000× *g* for 15 min [38]. The supernatant was retained to assay MDA content according to the method of Wang et al. [39]. Activities of superoxide dismutase (SOD) were determined as described by Pinhero et al. [40]. Catalase (CAT) and peroxidase (POD) activities were measured according to the methods of Wang et al. [41]. Proline content was measured according to the acid ninhydrin method described by Zhang et al. [42]. The activity of α-amylase was determined using the 3, 5-dinitrosalicylic acid colorimetric method [43]. Determination of each biochemical index included three biological replicates.

### 4.3. Determination of Phytohormone Content

Determination of phytohormone content was performed by Shanghai Sanshu Biotechnology Co., Ltd. (Shanghai, China) according to the method described by Liu et al. [44]. Briefly, 100 mg samples were ground to power in liquid nitrogen and extracted with 1.0 mL pre-chilled methanol: H_2_O: formic (7.9: 2: 0.1, *v*/*v*/*v*) mixture overnight at 4 °C. The supernatant was centrifuged at 13,000 rpm for 20 min at 4 °C, and the solid residue was re-extracted and recentrifuged. Pooled supernatants were passed through an Oasis MAX strong anion-exchange column (Waters, Milford, MA, USA) to remove interfering lipids and plant pigments and then dried under nitrogen gas. The residue was dissolved in 100 μL methanol and subjected to LC–MS/MS on an AB Sciex 5500 QTRAP spectrometer (AB Sciex, Toronto, ON, Canada). The LC–MS/MS was operated in the negative mode with the electrospray as the ionization source. Separation was performed on a Waters ACQUITY HSS T3 (100 mm × 2.1 mm, 1.8 μm) column. Gradient elution was applied with a mobile phase of methanol (A) and water (B) containing 0.1% formic acid at a flow rate of 0.3 mL/min. The column temperature was maintained at 40 °C, with an injection volume is 5 μL. The calibration standards included a mixed phytohormone standard solution containing zeatin, gibberellin 1 (GA_1_), gibberellin 4 (GA_4_), abscisic-acid (ABA), indole-acetic-acid (IAA) and jasmonic-acid (JA) standards (Sigma, New York, NY, USA). The calibration standards were prepared at concentrations of 0.1, 1, 5, 10, 20, 40, 60, 80 and 100 ng/mL for each phytohormone standard in the mixed standard solution of the six compounds. The content of each phytohormone was calculated based on the standard curves in units of ng per mg fresh weight (FW) using Analyst 1.6 software. All analyses were performed in three biological replicates.

### 4.4. RNA Isolation, cDNA Library Preparation and Transcriptome Sequencing

Total RNA of each sample was extracted using the RN38-EASYspin Plus Plant RNA Kit (Aidlab Biotechnology Co., Ltd., Beijing, China). The quality and quantity of total RNA was analyzed using a NanoDrop spectrophotometer (Thermo Fisher Scientific Inc., Waltham, MA, USA), and integrity was further evaluated using an Agilent 2100 Bioanalyzer (Agilent Technologies Co. Ltd., Santa Clara, CA, USA). High-quality RNA separated from two independent samples (biological replicates) was pooled for library preparation. Construction of the libraries and RNA-Seq was performed by CapitalBio Technology Corporation (Beijing, China), and the cDNA library was sequenced using Illumina HiSeq™2500.

### 4.5. Gene Quantification and Differential-Expression Analysis

Sequence reads were processed to remove adapter sequences, low-quality reads and very short-length reads. Paired reads were quality-filtered using NGS QC toolkit v 2.3 [45]. High-quality reads were used for de novo assembly using HISAT software [46] and aligned to the reference genome (Zea mays AGPv3.22, http://www.maizegdb.org/, 16 June 2020). The assembly resulted in contigs and singletons, together with a uniform set of unigenes. The abundance of unigene expression was calculated by estimating FPKM values (fragments per kilobase of transcript sequence per million sequenced base pairs). Analysis of differentially expressed genes (DEGs) among different treatments was performed with DESeq2 package [47], and DEGs were determined based on false-discovery rate (FDR values < 0.05) and fold change (|log2 (fold change) | ≥2). Venn diagrams and a cluster heat map of DEGs were generated in the bioinformatics platform (http://www.bioinformatics.com.cn, 19 September 2020).

### 4.6. GO and KEGG Pathway-Enrichment Analyses for Differentially Expressed Genes

Gene-ontology (GO) terms for DEGs were calculated using the AgriGO v2.0 toolkit [48], and the unigenes were assigned to biological functions on the macro levels of ‘biological process’, ‘cellular component’ and ‘molecular function’. KOBAS software was employed to analyze the statistical enrichment of DEG pathways in the Kyoto Encyclopedia of Genes and Genomes (KEGG) [49]. Enrichment analysis of DEGs was performed using Fisher’s exact test. Significant enrichment of the gene sets was detected with a corrected *p*-value < 0.05. Gene expression in the phenylpropanoid biosynthesis pathway was plotted by PATHVIEW [50].

### 4.7. Validation of RNA-Seq Data by Real-Time Quantitative PCR

Total RNA of each sample was extracted and converted to cDNA using PrimeScript RT reagent Kit with gDNA Eraser (TaKaRa, Dalian, China). Real-time quantitative PCR (qRT-PCR) was performed in 96-well plates with a Bio-Rad CFX96 Detection System (Bio-Rad) using the SYBR premix EX Taq (TaKaRa). Primer sequences designed with Primer Premier 5.0 software are shown in Appendix A. Gene expression was calculated in relation to the reference gene *β-TUB* using the 2^−ΔΔCT^ method [51]. At least three independent biological replicates for each sample and three technical replicates of each biological replicate were arranged to ensure the reproducibility and reliability of qRT-PCR results.

### 4.8. Quantitative Real-Time PCR Expression Analysis of Candidate Genes

Dry seeds of Mo17 and B73 were surface-sterilized with 0.5% sodium hypochlorite solution for 5 min, followed by thorough washing with water. Then, seeds were soaked in tap water for 1 d at 25 °C. Next, seeds were sowed in moist sand with a depot of 2 or 20 cm. All seeds were germinated in growth chambers with a photosynthetic photon-flux density (PPFD) of 250 μmol m^−2^s^−1^ and a 12-h photoperiod at 25 °C. Embryos were rapidly separated from seeds germinated for 2 d. Total RNA was extracted from separated embryos and converted to cDNA using PrimeScript RT reagent Kit with gDNA Eraser (TaKaRa, Dalian, China) following the manufacturer’s instructions. Real-time quantitative PCR was conducted in Bio-Rad CFX96™ (Bio-Rad, USA) using SYBR premix EX Taq (TaKaRa). Gene-specific primers were designed using Primer 5.0. With maize *Zmβ-TUB* as the reference gene, relative gene-expression values were calculated using the 2^−ΔΔCT^ method [51]. The gene-specific primers are listed in Appendix A.

### 4.9. Statistical Analysis

Obtained metabolite data were statistically analyzed using one-way analysis of variance (ANOVA), which was carried out with the PROC GLM procedure in Statistical Analysis System (SAS) software using the least significant difference test at the 0.05% level of significance. Before analysis, the percentage data were transformed according to ŷ = arcsin [sqr (x/l00)].

## 5. Conclusions

In conclusion, some key biochemical pathways and candidate genes closely related to maize deep sowing were screened in this study. The selected candidate genes were further verified in maize inbred lines with varying deep-sowing tolerance (Mo17 and B73). Our results suggest that although the molecular mechanism of maize deep-sowing germination is very complex and still unclear, phenylpropanoid biosynthesis, starch and sucrose metabolism, response to oxidative stress and antagonistic or synergistic regulation between phytohormones, such as GA_4_, IAA, zeatin, ABA and JA, played crucial roles during the seed germination process. This study provides a theoretical basis for further clarification of the complex regulation mechanism of deep-sowing tolerance of maize and the breeding of deep-sowing-tolerant maize varieties.

## Figures and Tables

**Figure 1 plants-11-00359-f001:**
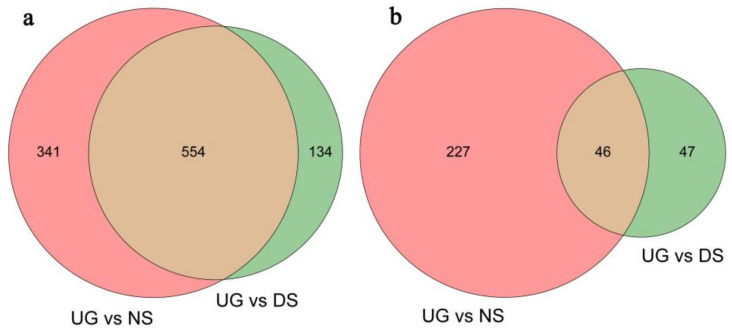
Comparative analysis of transcriptome changes in maize embryos in response to different sowing conditions. Venn diagram constructed using the DEGs with changes of more than two folds and FDR values of less than 0.05. (**a**,**b**) Number of upregulated and downregulated DEGs, respectively, related to germination.

**Figure 2 plants-11-00359-f002:**
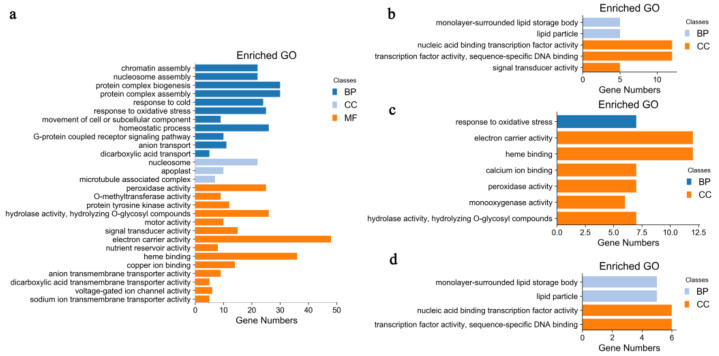
Gene-ontology functional-enrichment analysis of DEGs detected in maize embryos. (**a**,**b**) Significant GO terms associated with up- and downregulated DEGs, respectively, detected between DS and UG. (**c**,**d**) Significant GO terms associated with up- and downregulated DEGs, respectively, detected only in DS.

**Figure 3 plants-11-00359-f003:**
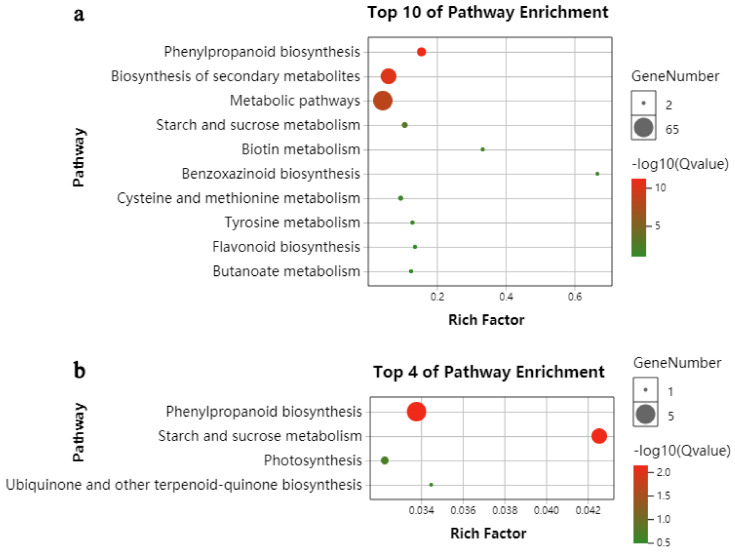
Bubble chart of the enriched KEGG pathway in upregulated DEGs. (**a**,**b**) Bubble chart of the enriched KEGG pathway detected between DS and UG and in maize embryos of DS, respectively.

**Figure 4 plants-11-00359-f004:**
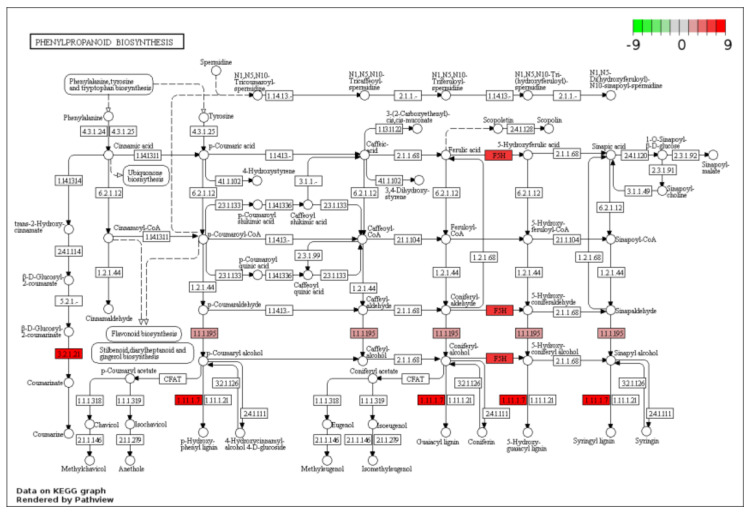
Gene expression in the phenylpropanoid biosynthesis pathway plotted by PATHVIEW. Red represents upregulated genes, while green represents downregulated genes. The depth of color indicates the relative expression level of these genes.

**Figure 5 plants-11-00359-f005:**
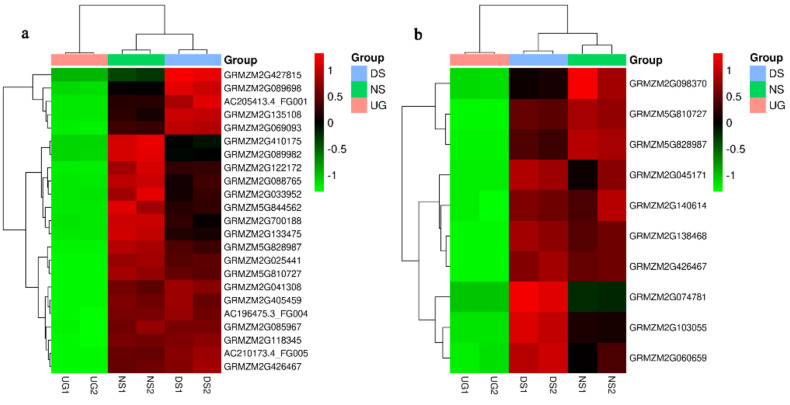
Detailed expression profile of DEGs related to the phenylpropanoid biosynthesis pathway (**a**) and the starch and sucrose metabolism pathway (**b**) based on KEGG analysis in maize embryos. Heat maps illustrate the relative expression level of these genes.

**Figure 6 plants-11-00359-f006:**
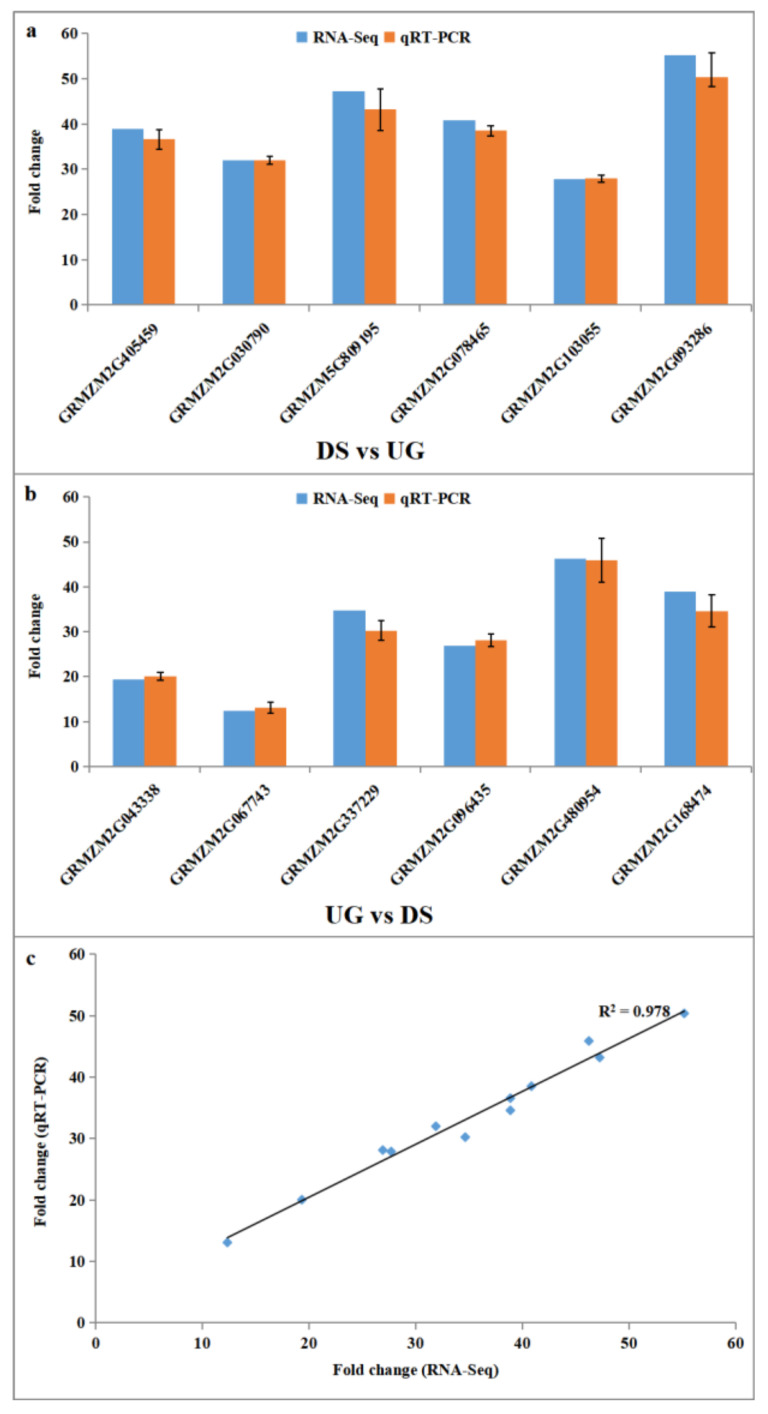
Real-time quantitative PCR (qRT-PCR) validation of RNA-Seq results. (**a**,**b**) Expression of six upregulated DEGs and six downregulated DEGs validated by qRT-PCR and compared with their expression obtained from RNA-Seq. (**c**) Pearson correlation analysis of gene-expression ratios obtained from qRT-PCR and RNA-Seq data. Error bars indicate standard error of the mean.

**Figure 7 plants-11-00359-f007:**
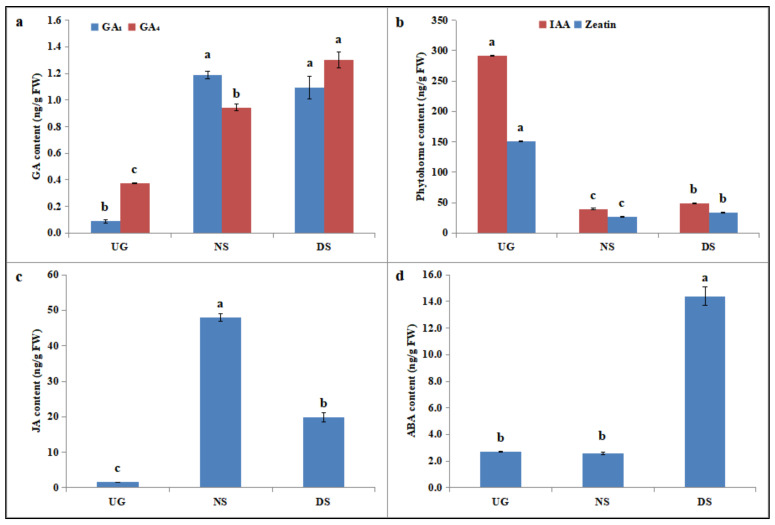
Effect of deep sowing on phytohormone changes in maize embryos. Means with standard deviations not followed by the same lower-case letter significantly differ by Tukey test at a 5% level of significance. (**a**) Changes in gibberellin 1 and 4 levels in maize embryos; (**b**) changes in indole acetic-acid and zeatin levels in maize embryos; (**c**,**d**) changes in jasmonic-acid and abscisic-acid levels in maize embryos, respectively.

**Table 1 plants-11-00359-t001:** Effect of deep-sowing on physiological changes in maize embryos.

Treatments	SOD (U/g FW)	CAT (U/g FW)	POD (U/g FW)	MDA (μmol/L)	Pro (μg/g FW)	α-Amylase (mg/(g·min))
UG	128.83 ± 0.61 c ^a^	6.00 ± 0.06 c	62.5 ± 12.5 c	0.16 ± 0.01 c	5.63 ± 0.03 b	10.54 ± 0.75 c
NS	155.69 ± 3.34 b	8.94 ± 0.25 b	125.1 ± 10.1 b	0.19 ± 0.01 b	5.89 ± 0.01 b	54.84 ± 2.43 b
DS	201.50 ± 0.60 a	15.56 ± 0.06 a	225.2 ± 25 a	0.36 ± 0.02 a	13.59 ± 0.11 a	103.71 ± 5.49 a

^a^ Means with standard deviations not followed by the same lower-case letter significantly differ by LSD test at a 5% level of significance.

**Table 2 plants-11-00359-t002:** Effect of deep sowing on candidate-gene expression in different deep-sowing-tolerant maize inbred lines.

Inbred Line	GRMZM2G103055	GRMZM5G809195	GRMZM2G030790	GRMZM2G405459	GRMZM2G043338	GRMZM2G337229
B73	5.18 ± 0.14 a ^a^	5.5 ± 0.34 a	11.71 ± 1.06 a	11.09 ± 0.31 a	3.44 ± 0.28 a	7.55 ± 0.97 a
Mo17	2.69 ± 0.08 b	1.12 ± 0.02 b	10.69 ± 0.26 a	1.65 ± 0.01 b	0.52 ± 0.03 b	0.39 ± 0.04 b

^a^ Different letters after the standard deviation indicate significant differences between B73 and Mo17 at a significance level of *p* < 0.05, according to the LSD test.

## Data Availability

Not applicable.

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
