# Peer review of "Transcriptome Analysis Revealed the Key Genes and Pathways Involved in Seed Germination of Maize Tolerant to Deep-Sowing"

_plants, 2022, doi:10.3390/plants11030359_

Round 1

Reviewer 1 Report

The study seems to set out and characterise a genetic mechanism that underlies the germination of maize seeds tolerant to deep sowing. For this, authors have used molecular biology and biochemical techniques comparing embryo samples from seeds that: did not germinate, germinated under 2 cm sowing; and germinated under 20 cm sowing. In the conclusion section, authors claim that “phenylpropanoid biosynthesis, starch and sucrose metabolism, response to oxidative stress and antagonistic or synergistic regulation between phytohormones played crucial roles”.

I have two major concerns regarding the current narrative of this manuscript:

Firstly, seed germination and seedling establishment are well-defined processes, with the first one ending when the radicle protrudes through the surrounding tissues and the second one starting thereafter. Thus, comparing germinated with ungerminated seeds creates a somewhat confusing narrative. For example, lignin synthesis, which require phenylpropanoids, and starch mobilization are known processes involved in seedling establishment. Hence, it can be easily argued that the differential expression of observed genes is not the basis of germination, but its consequence given that germinated seeds will be entering a new developmental phase. This is a major obstacle in my view, so it is essential to clearly define these events prior to attributing a particular mechanism or measurement to them.

Secondly, and possible due to the abovementioned concern, the manuscript lacks a clear objective and conclusion. It is vague what authors expect to understand from the implemented experimental design. Is it germination stricto sensu, or aspects of seedling establishment such as hypocotyl elongation, seedling emergence? In the conclusion section (lines 534-544), authors spend half of the paragraph describing execution of the experiments as opposed to asserting its inferences. Overall, the authors could improve communication of their work so that outputs are articulated with more care and precision.

Author Response

Dear reviewer,

Thank you for giving us the opportunity to revise our article entitled “Transcriptome analysis revealed the key genes and pathways involved in seed germination of maize tolerant to deep-sowing” (plants-1543925). The reviewer raised some good questions or suggestions for our paper. Now, we answer them one by one as follows.

Reviewer 1:

The study seems to set out and characterise a genetic mechanism that underlies the germination of maize seeds tolerant to deep sowing. For this, authors have used molecular biology and biochemical techniques comparing embryo samples from seeds that: did not germinate, germinated under 2 cm sowing; and germinated under 20 cm sowing. In the conclusion section, authors claim that “phenylpropanoid biosynthesis, starch and sucrose metabolism, response to oxidative stress and antagonistic or synergistic regulation between phytohormones played crucial roles”.

I have two major concerns regarding the current narrative of this manuscript:

Question 1: Firstly, seed germination and seedling establishment are well-defined processes, with the first one ending when the radicle protrudes through the surrounding tissues and the second one starting thereafter. Thus, comparing germinated with ungerminated seeds creates a somewhat confusing narrative. For example, lignin synthesis, which require phenylpropanoids, and starch mobilization are known processes involved in seedling establishment. Hence, it can be easily argued that the differential expression of observed genes is not the basis of germination, but its consequence given that germinated seeds will be entering a new developmental phase. This is a major obstacle in my view, so it is essential to clearly define these events prior to attributing a particular mechanism or measurement to them.

Reply: For this question, I think it can be understood that the purpose of our study was to find out some genes or metabolic pathways related to deep sowing tolerance of maize seeds. Therefore, we first take the maize seeds soaked in tap water for 1 day (ungerminated seeds, UG) as CK1, then we compared the seeds germinated for 2 days under normal sowing condition (NS, which can also be considered as CK2) with CK1 to find out which genes or metabolic pathways were regulated during maize seed germination under normal sowing; secondly, we compared the seeds germinated for 2 days under deep sowing condition (DS) with UG to find out genes or pathways regulated during seed germination under deep sowing; finally, we compared and analyzed the results of the above two comparisons to find out the specifically expressed genes or metabolic pathways induced by deep-sowing. These results were the final goal of this study. Consequently, we believed that there should be no major problems in our experimental design.

Question 2: Secondly, and possible due to the above mentioned concern, the manuscript lacks a clear objective and conclusion. It is vague what authors expect to understand from the implemented experimental design. Is it germination stricto sensu, or aspects of seedling establishment such as hypocotyl elongation, seedling emergence? In the conclusion section (lines 534-544), authors spend half of the paragraph describing execution of the experiments as opposed to asserting its inferences. Overall, the authors could improve communication of their work so that outputs are articulated with more care and precision.

Reply: We have deleted the experimental description in the conclusion section (lines 544-545 in the revised manuscript) and modified “the conclusion section” (lines 550-552 in the revised manuscript) to make it more refined.

Reviewer 2 Report

It is a quite well-written manuscript. Undoubtedly, the experiments were well planned, and the results are well presented and described. Generally, I have no substantive objections to the text, so I recommend the acceptance of the manuscript. However, a minor correction is needed.

  • A piece of information about experiments repetitions must be added. It is crucial for RNA-Sec. How many times was this analysis repeated? Three repetitions are the best.
  • Keywords should be changed because keywords should not be the same as words already used in a title.
  • Figure S1 is referred to in line 106, and in this place is explained what this figure is showing. That is ok, but the figure's legend should also be added just under Figure S1.
  • I encourage the authors to present Table 1 and Table 2 as figures.
  • Sections 4.3 and 4.8. Superscripts and subscripts must be corrected. H2O or Celsius degree.

Author Response

Dear  reviewer,

Thank you for giving us the opportunity to revise our article entitled “Transcriptome analysis revealed the key genes and pathways involved in seed germination of maize tolerant to deep-sowing” (plants-1543925). The editors and reviewers raised some good questions or suggestions for our paper. Now, we answer them one by one as follows.

Reviewer 2:

It is a quite well-written manuscript. Undoubtedly, the experiments were well planned, and the results are well presented and described. Generally, I have no substantive objections to the text, so I recommend the acceptance of the manuscript. However, a minor correction is needed.

Question 1: A piece of information about experiments repetitions must be added. It is crucial for RNA-Sec. How many times was this analysis repeated? Three repetitions are the best.

Reply: Line 488-490 in the revised manuscript had described that the analysis of RNA-Seq was based on two biological repetitions.

Question 2: Keywords should be changed because keywords should not be the same as words already used in a title.

Reply: We have changed the keywords in our revised manuscript (see line 33-34).

Question 3: Figure S1 is referred to in line 106, and in this place is explained what this figure is showing. That is ok, but the figure's legend should also be added just under Figure S1.

Reply: We have added the figure's legend at the bottom of Figure S1 (see the revised Figure S1 in the supplementary file).

Question 4: I encourage the authors to present Table 1 and Table 2 as figures.

Reply: Combining the comments of Reviewer 2 and Reviewer 3, we retained the Tables, as its may more direct and clearer to show the differences between experiment treatments by using specific numerical numbers.

Question 5: Sections 4.3 and 4.8. Superscripts and subscripts must be corrected. H2O or Celsius degree.

Reply: We have made corresponding amendments as the reviewer’s suggestion (see line 464, 465 and 529 in the revised manuscript).

Reviewer 3 Report

I have read, carefully and with great interest, the manuscript entitled „Transcriptome analysis revealed the key genes and pathways involved in seed germination of maize tolerant to deep-sowing” prepared by Wang et al., for publication in Plants MDPI.

The aim of this study was to report and provide, for the first time, insights into the molecular mechanisms of maize germination as resulted by deep-sowing.

In my opinion the manuscript is of high relevance and will be interested for Plants readers. The manuscript is very good prepared and clearly described. The authors performed many analysis. The description of the research is correct and very detailed.

However some corrections/changes will improve its quality before final publication.

Line 19 and 21 – Explain the meanings of abbreviations „GO” and „KEGG”, they are provided in the abstract for the first time.

Keywords – Avoid using words from manuscript title as keywords (don’t repeat them, use other instead to broaden the information about your manuscript).

The names „cultivar” and „variety” should not be used interchangeably, their meanings are different.

Figure 7 – I think that „CK” is used instead of „UG” on the x axis legend.

Table 2 – Footnote – I suggest to delete the methodological informations.

Materials and methods:

- subsection 4.1 – How many seeds were used in each experimental object? This information should be added.

- subsection 4.2 – How many times were the biochemical analysis repeated?

- subsection 4.9 – What specific LSD test was used? Add information.

Author Response

Dear  reviewer,

Thank you for giving us the opportunity to revise our article entitled “Transcriptome analysis revealed the key genes and pathways involved in seed germination of maize tolerant to deep-sowing” (plants-1543925). The reviewer raised some good questions or suggestions for our paper. Now, we answer them one by one as follows.

Reviewer 3:

I have read, carefully and with great interest, the manuscript entitled “Transcriptome analysis revealed the key genes and pathways involved in seed germination of maize tolerant to deep-sowing” prepared by Wang et al., for publication in Plants MDPI.

The aim of this study was to report and provide, for the first time, insights into the molecular mechanisms of maize germination as resulted by deep-sowing.

In my opinion the manuscript is of high relevance and will be interested for Plants readers. The manuscript is very good prepared and clearly described. The authors performed many analysis. The description of the research is correct and very detailed.

However some corrections/changes will improve its quality before final publication.

Question 1: Line 19 and 21-Explain the meanings of abbreviations “GO” and “KEGG”, they are provided in the abstract for the first time.

Reply: We have added the meaning of abbreviations of “GO” and “KEGG” (see line 20 and 23 in the revised manuscript).

Question 2: Keywords-Avoid using words from manuscript title as keywords (don’t repeat them, use other instead to broaden the information about your manuscript).

Reply: We have changed the keywords in our revised manuscript (see line 33-34).

Question 3: The names “cultivar” and “variety” should not be used interchangeably, their meanings are different.

Reply: We have changed “cultivar” to “variety” (see line 47 and 324 in the revised manuscript).

Question 4: Figure 7-I think that “CK” is used instead of “UG” on the x axis legend.

Reply: We are sorry for the mistake, and we have changed “CK” to “UG” in Figure 7 (see the revised Figure 7 in the manuscript figures).

Question 5: Table 2-Footnote-I suggest to delete the methodological informations.

Reply: We have deleted the methodological informations of Table 2 as the reviewer’s suggestion (see line 305-306 in the revised manuscript).

Question 6: subsection 4.1-How many seeds were used in each experimental object? This information should be added.

Reply: We have added the information, i.e., “Each treatment contained three repetitions with 100 seeds per repeat.” (see line 442-443 in the revised manuscript).

Question 7: subsection 4.2-How many times were the biochemical analysis repeated?

Reply: We have added the information, i.e., “The determination of each biochemical index included three biological replicates.” (see line 458-459 in the revised manuscript).

Question 8: subsection 4.9-What specific LSD test was used? Add information.

Reply: We have added the information of which specific LSD test was used in our study (see line 538-539 in the revised manuscript).

Round 2

Reviewer 1 Report

I believe my concern still stands and I reiterate that it is critical that authors define which developmental stage is being probed prior to attributing a particular mechanism or measurement to it. Thus, it is my understanding that the data does not answer questions concerning the control of seed germination per se for reasons mentioned in the first review. Authors now say, “I think it can be understood that the purpose of our study was to find out some genes or metabolic pathways related to deep sowing tolerance of maize seeds”. However, instead of a general understanding of the genes and/or metabolic pathways underlying deep sowing tolerance in maize, which evidently involves seed germination and seedling establishment, the manuscript is very specific in asking questions and claiming answers to the control of seed germination. Here are a few examples:

In the title, authors state that “Transcriptome analysis revealed the key genes and pathways involved in seed germination of maize tolerant to deep-sowing”.

In lines 16-19, authors begin with “To improve our understanding of maize seed germination mechanismunder deep sowing…”.

In lines 29 and 30, authors claim “These findings enriched our knowledge about the key biochemical pathways and genes regulating maize seed germination…”.

In lines 74 and 75, authors say “…making it possible to better understand the molecular mechanism of seed germination in response to deep-sowing…”.

In lines 336 and 336, “In order to reveal the molecular mechanism of maize seed germination under deep-sowing condition…”.

In lines 538-542, authors conclude that “Our results suggested that although the molecular mechanism of maize deep sowing germination is very complex and still unclear, phenylpropanoid biosynthesis, starch and sucrose metabolism, response to oxidative stress and antagonistic or synergistic regulation between phytohormones, such as GA4, IAA, zeactin, ABA and JA, played crucial roles during the seed germination process”.

As it stands, the current data does not support the conclusions made in the manuscript.